# The first 2 months of the SARS-CoV-2 epidemic in Yemen: Analysis of the surveillance data

Ali Ahmed Al-Waleedi[1], Jeremias D. Naiene[2]*, Ahmed A. K. Thabet[2], Adham Dandarawe[1], Hanan Salem[1], Nagat Mohammed[1], Maysa Al Noban[1], Nasreen Salem Bin-Azoon[1], Ammar Shawqi[1], Mohammed Rajamanar[1], Riyadh Al-Jariri[1†], Mansoor Al Hyubaishi[1], Lina Khanbari[2], Najib Thabit[2], Basel Obaid[2], Manal Baaees[2], Denise Assaf[2], Mikiko Senga[2], Ismail Mahat Bashir[2], Nuha Mahmoud[2], Roy Cosico[2], Philip Smith[2], Altaf Musani[2]

1 Ministry of Public Health and Population, Aden, Yemen, 2 World Health Organization, Sana'a, Yemen

† Deceased.
* naienej@who.int

**Data Availability Statement:** All relevant data are within the manuscript and its Supporting Information files.

## Abstract

### Introduction

Yemen was one of the last countries in the world to declare the first case of the pandemic, on 10 April 2020. Fear and concerns of catastrophic outcomes of the epidemic in Yemen were immediately raised, as the country is facing a complex humanitarian crisis. The purpose of this report is to describe the epidemiological situation in Yemen during the first 2 months of the SARS-CoV-2 epidemic.

### Methods

We analyzed the epidemiological data from 18 February to 05 June 2020, including the 2 months before the confirmation of the first case. We included in our analysis the data from 10 out of 23 governorates of Yemen, located in southern and eastern part of the country.

### Results

A total of 469 laboratory confirmed, 552 probable and 55 suspected cases with onset of symptoms between 18 February and 5 June 2020 were reported through the surveillance system. The median age among confirmed cases was 46 years (range: 1–90 years), and 75% of the confirmed cases were male. A total of 111 deaths were reported among those with confirmed infection. The mean age among those who died was 53 years (range: 14–88 years), with 63% of deaths (n = 70) occurring in individuals under the age 60 years. A total of 268 individuals with confirmed SARS-CoV-2 infection were hospitalized (57%), among whom there were 95 in-hospital deaths,

### Conclusions

The surveillance strategy implemented in the first 2 months of the SARS CoV 2 in the southern and eastern governorates of Yemen, captured mainly severe cases. The mild and

**Funding:** The author received no specific funding for this work.

**Competing interests:** The authors have declared that no competing interests exist.

moderate cases were not self-reported to the health facilities and surveillance system due to limited resources, stigma, and other barriers. The mortality appeared to be higher in individuals aged under 60 years, and most fatalities occurred in individuals who were in critical condition when they reached the health facilities. It is unclear whether the presence of other acute comorbidities contributed to the high death rate among SARS-CoV-2 cases. The findings only include the southern and eastern part of the country, which is home to 31% of the total population of Yemen, as the data from the northern part of the country was inaccessible for analysis. This makes our results not generalizable to the rest of the country.

## Introduction

Although pneumonia caused by a novel coronavirus was first described in China in December 2019, it was officially named as coronavirus disease 2019 (COVID-19) on 11 February 2020 [1]. The virus that causes the disease was named severe acute respiratory syndrome coronavirus-2 (SARS-CoV-2) due to its genetic similarities to the coronavirus responsible for the severe acute respiratory syndrome (SARS) epidemic in 2003 [2]. SARS-CoV-2 is transmitted through respiratory droplets and the case fatality rate ranges from 0.3% to 15% among the confirmed cases, primarily due to pulmonary complications [1, 3]. The incubation period is estimated to be from 2 to 14 days, with majority of the cases developing symptoms 5 days after exposure to the virus. The signs and symptoms may include fever, cough, difficult breathing, fatigue, headache and others [4]. The virus can also be transmitted by individuals who are asymptomatic carriers of the virus [5]. The development of symptoms and fatality varies by age group, with older people being most at risk of becoming symptomatic and die [6].

Yemen was one of the last countries in the world to declare the first case of the pandemic, on 10 April 2020 [7]. Fear and concerns of catastrophic outcomes of the epidemic in Yemen were immediately raised, as the country is facing a complex humanitarian crisis [7]. Suppression measures were already being implemented by the government before the notification of the first case, including curfew, closure of schools, airports, markets, mosques, and prohibiting public gatherings [8].

About 50% of the population of Yemen is estimated to be in acute need of health care, with high rates of malnutrition, child and maternal mortality [7]. In addition, the limited availability of safe drinking water, people living crowded houses, inadequate sanitation, and stigma constitute barriers for effective response and control of the epidemic in Yemen [9]. Due to the ongoing armed conflict, less than 50% of the health facilities in Yemen are fully functional [7] and more that 2 million children are malnourished [7]. Yemen has a population of 30 million people, and the armed conflict which started in 2015 led to a fragmentation of the healthcare system [7, 8].

For early detection of SARS-CoV-2 in Yemen, as in other countries, a case definition, active surveillance, and contact tracing were required [10, 11]. The purpose of this report is to describe the epidemiological situation in Yemen during the first 2 months of the SARS-CoV-2 epidemic, from 5 April to 5 June 2020. The report also includes the 2 months before the notification of the first confirmed case from 18 February to 10 April 2020, as well as the challenges and information gaps.

## Material and methods

### Data source and analysis

We analyzed the epidemiological data from 18 February to 05 June 2020. We included in our analysis the data from 10 out of 23 governorates of Yemen, namely Abyan, Aden, Al-Dhale'e,

Al-Mahrah, Hadramout, Lahj, Marib, Shabwa, Taizz and the island of Socotra, located in southern and eastern part of the country controlled by the internationally recognized government. Yemen has a total population of 30 million according to projections from the 2004 census, with 31% of the population located in the southern and eastern governorates. The data from the northern governorates were not available for analysis. The cases and contacts were investigated by 5-member multidisciplinary rapid response teams (RRTs) in each district, comprising a clinician, laboratory technician, surveillance officer, risk communication officer and an environmental health officer. A line list of cases and contacts was compiled daily in a Microsoft Excel spreadsheet using the data received from all the governorates. All the variables in the line list were extracted from the case-based form used in the country. These included demographic information, signs and symptoms, history of contact with other cases, history of travel, comorbidities, and hospitalization data. The comorbidities only included some specific non-communicable diseases, although information regarding communicable diseases and other health conditions was sometimes recorded in the comments section of the form. Contact tracing data were available only from Hadramout governorate. We constructed the chains of transmission of the other governorates using the information available in the line list. The contact tracing activities started with the reporting of the first confirmed case, who had onset of illness on 5 April 2020. We decided to limit the analysis of the chains of transmission to the first 250 cases, with onset of illness from 5 April to 25 May 2020, when the collection of the information was more consistent. The contacts were followed up daily by the RRTs. Each team was responsible to follow up a maximum of 10 contacts to ensure quality and consistence of contact tracing data. There was no specialized software available for contact tracing and the daily analysis was done through Microsoft Excel. A temporary interruption followed by inconsistency of contact tracing data reporting was observed after 25 May 2020. When the number of contacts exceeded the recommended 10 contacts per contact tracing team the daily information and analysis regarding the status of the contacts became irregular. We used the Microsoft Excel to perform univariate analysis of the cases and contacts data.

## Case definition

Yemen adopted the WHO case definition of suspected, probable, and confirmed cases in March 2020 [12] (Fig 1). The suspected cases had to present acute respiratory illness and were subdivided in the following components: a component "A" with history of travel to affected countries, a component "B" with history of contact with confirmed or probable cases and a

**Suspect case**

A. A patient with acute respiratory illness (fever and at least one sign/symptom of respiratory disease, e.g., cough, shortness of breath), AND a history of travel to or residence in a location reporting community transmission of COVID-19 disease during the 14 days prior to symptom onset;

OR

B. A patient with any acute respiratory illness AND having been in contact with a confirmed or probable COVID-19 case (see definition of contact) in the last 14 days prior to symptom onset;

OR

C. A patient with severe acute respiratory illness (fever and at least one sign/symptom of respiratory disease, e.g., cough, shortness of breath; AND requiring hospitalization) AND in the absence of an alternative diagnosis that fully explains the clinical presentation.

**Probable case**

A. A suspect case for whom testing for the COVID-19 virus is inconclusive.

OR

B. A suspect case for whom testing could not be performed for any reason.

**Confirmed case**

A person with laboratory confirmation of COVID-19 infection, irrespective of clinical signs and symptoms.

**Fig 1. World Health Organization SARS-CoV-2 case definition adopted by Yemen in March 2020.**

component "C" requiring hospitalization, without a clear diagnosis, regardless to history of travel of contact with sick people. Due to limited laboratory supplies, component "C" of definition of a suspected case was given low priority for several weeks. Probable cases were suspected cases either not tested or with inconclusive laboratory results. Cases with positive laboratory results for COVID-19 were classified as confirmed.

## Laboratory confirmation

Real-time reverse transcriptase polymerase chain reaction (RT-PCR) performed to nasopharyngeal swabs specimens was used for laboratory confirmation of SARS-CoV-2 infection. The RRTs transported the specimens to the laboratories in viral transport medium. Only cases that were positive for SARS-CoV-2 using RT-PCR were classified as confirmed. A total of 5 central public health laboratories (CPHLs) were available in different geographical areas of the country to perform RT-PCR, namely in Aden, Mukalla, Sayoun, Taizz, and Sana'a. Data for the CPHL in Sana'a were not provided by the local authorities from the north of Yemen, and therefore were not included in our analysis. The Aden CPHL started SARS-CoV-2 testing on 21 March 2020, and confirmed the first case on 29 April 2020. Mukalla CPHL started SARS-CoV-2 testing on 8 April 2020, and confirmed the first case confirmed on 10 April 2020. This was the first confirmed case in Yemen. The Taizz CPHL starting testing for SARS-CoV-2 on 27 April, and confirmed the first case on 1 May 2020. Sayoun was the last CPHL to start testing for SARS-CoV-2 on 6 May 2020, and confirmed the first case on 9 May 2020.

## Ethical considerations

The analysis and publication of the data was authorized by the scientific committee of the Ministry of Public Health and Population in Aden. No additional ethical approval was required because the data collection and analysis were part of an outbreak response and not a research. The requirement for informed consent was waived because the study is based on a retrospective analysis of routine surveillance data collected in an emergency. The patients and health workers details were kept confidential.

## Results

### Demographic information

A total of 469 laboratory confirmed, 552 probable and 55 suspected cases with onset of symptoms between 18 February and 5 June 2020 were reported through the surveillance system (Table 1). Of the confirmed cases, 3 had history of travel abroad within 14 days before the

**Table 1. Distribution of suspected, probable, and confirmed cases of SARS-CoV-2 infection in Yemen according to governorate, 18 February to 5 June 2020.**

| Governorate | Confirmed | | Probable | | Suspected | |
|---|---|---|---|---|---|---|
| | Cases | Deaths | Cases | Deaths | Cases | Deaths |
| Abyan | 14 | 2 | 38 | 16 | 2 | 0 |
| Aden | 126 | 5 | 89 | 10 | 0 | 0 |
| Al Dhale'e | 15 | 4 | 91 | 26 | 12 | 1 |
| Al Maharah | 3 | 1 | 0 | 0 | 1 | 0 |
| Hadramaut (Al-Mukalla) | 94 | 40 | 0 | 0 | 0 | 0 |
| Hadramaut (Say'on) | 32 | 8 | 30 | 3 | 7 | 0 |
| Lahj | 50 | 17 | 181 | 34 | 8 | 0 |
| Marib | 13 | 4 | 105 | 9 | 0 | 0 |
| Shabwah | 24 | 6 | 11 | 4 | 23 | 4 |
| Taizz | 98 | 24 | 7 | 0 | 2 | 0 |
| **Total** | **469** | **111** | **552** | **102** | **55** | **5** |

onset of symptoms, namely to Saudi Arabia (n = 2) and Egypt (n = 1). The median age among confirmed cases was 46 years (range: 1–90 years), and 75% of the confirmed cases were male (Table 2).

Three confirmed cases were reported in children under 5 years old. A total of 111 deaths were reported among those with confirmed infection, 71% of them (n = 79) occurring in males. The mean age among those who died was 53 years (range: 14–88 years), with 63% of deaths (n = 70) occurring in individuals under the age 60 years. Hadramout governorate reported the highest number of deaths (n = 40). Among those with confirmed infection, diabetes (11%) was the most common comorbidity among those with confirmed infection, and

**Table 2. Demographic characteristics of individuals with confirmed SARS-CoV-2 infection in Yemen from 18 February to 5 June 2020.**

|  | Cases | Deaths | Recovered |
|---|---|---|---|
|  | N (%) | N (%) | N (%) |
| Overall | 469 | 111 | 23 |
| Age (years) |  |  |  |
| 0–9 | 3 (1) | 0 (0) | 0 (0) |
| 10–19 | 6 (1) | 1 (1) | 1 (4) |
| 20–29 | 51 (11) | 5 (5) | 3 (13) |
| 30–39 | 88 (19) | 8 (7) | 8 (35) |
| 40–49 | 118 (25) | 29 (26) | 7 (30) |
| 50–59 | 89 (19) | 27 (24) | 3 (13) |
| 60–69 | 71 (15) | 22 (20) | 1 (4) |
| 70–79 | 29 (6) | 11 (10) | 0 (0) |
| $\geq$80 | 14 (3) | 8 (7) | 0 (0) |
| Sex |  |  |  |
| Female | 118 (25) | 32 (29) | 4 (17) |
| Male | 351 (75) | 79 (71) | 19 (83) |
| Comorbid Conditions |  |  |  |
| Cardiovascular Disease | 29 (6) | 9 (8) | 0 (0) |
| Hypertension | 38 (8) | 19 (17) | 2 (9) |
| Chronic Lung Disease | 23 (5) | 14 (13) | 1 (4) |
| Diabetes | 52 (11) | 18 (16) | 3 (13) |
| Kidney disease | 10 (2) | 4 (4) | 0 (0) |
| Liver disease | 7 (1) | 3 (3) | 1 (4) |
| Pregnancy | 1 (0) | 0 (0) | 1 (4) |
| HIV | 1 (0) | 1 (1) | 0 (0) |
| Others | 24 (5) | 10 (9) | 3 (13) |
| No underlying conditions | 284 (61) | 33 (30) | 12 (52) |
| Governorate |  |  |  |
| Abyan | 14 (2) | 2 (2) | 4 (17) |
| Aden | 126 (5) | 5 (5) | 0 (0) |
| Al Dhale'e | 15 (4) | 4 (4) | 0 (0) |
| Al Maharah | 3 (1) | 1 (1) | 1 (4) |
| Hadramaut (Al-Mukalla) | 94 (40) | 40 (36) | 7 (30) |
| Hadramaut (Say'on) | 32 (8) | 8 (7) | 5 (22) |
| Lahj | 50 (17) | 17 (15) | 2 (9) |
| Marib | 13 (4) | 4 (4) | 2 (9) |
| Shabwah | 24 (6) | 6 (5) | 0 (0) |
| Taizz | 98 (24) | 24 (22) | 2 (9) |

**Table 3.** The most common signs and symptoms among individuals with confirmed SARS-CoV-2 infection in Yemen from 18 February to 5 June 2020 (N = 469).

| Symptom or sign | n | % |
|---|---|---|
| Fever | 427 | 91 |
| Sore throat | 288 | 61 |
| Difficult breathing | 289 | 62 |
| Muscle and joint pain | 247 | 53 |
| Running nose | 113 | 24 |
| Cough | 401 | 86 |

hypertension (17%), diabetes (16%), and chronic lung diseases (13%) were the most common comorbidities among those who died. One of the fatal cases from Hadramout governorate was in an individual with HIV infection. A pregnant woman of approximately 20 weeks gestation was reported in Hadramout. Her pregnancy ended in a spontaneous miscarriage after the onset of her COVID-19 symptoms. No underlying conditions were reported in 61% of the cases.

The most common symptoms were fever, cough, and difficulty breathing and sore throat (Table 3). In addition to the symptoms shown in the table, 26 individuals reported a headache and 6 reported a loss of smell and taste.

Housewives were the most commonly affected occupational category, followed by healthcare workers and soldiers (Table 4). Among the healthcare workers, medical doctors and laboratory technicians were the most frequently affected, with 3 deaths of medical doctors and 2 deaths of laboratory technicians reported.

A marked increase occurred in number of cases reported according to the date of onset of symptoms from 24 April 2020, with a peak on 15 May 2020 (Fig 2). Aden and Hadramout were the governorates that reported the highest numbers of confirmed cases, followed by Taizz. Although the first case that was confirmed in Aden tested on 29 April 2020, 20 other cases in the governorate were subsequently confirmed that had earlier dates of onset of symptoms than the first case.

A total of 268 individuals with confirmed SARS-CoV-2 infection were hospitalized (57%), among whom there were 95 in-hospital deaths, and 27 were admitted to an intensive care unit (ICU), including 14 of those who died subsequently. Of the patients admitted to an ICU, 11 were provided with mechanical ventilation, including 8 of those who died (Table 5). Among the 95 patients who died in hospital, 44 had both the date of admission and the date of death

**Table 4.** Status according to occupational category among the 6 most common occupational categories among those with confirmed SARS-CoV-2 infection in Yemen, 18 February to 5 June 2020.

| Occupation | Cases | Deaths | Recovered | Under treatment |
|---|---|---|---|---|
| Housewives | 38 | 13 | 4 | 21 |
| Healthcare workers | 37 | 8 | 0 | 29 |
| *Doctors* | 18 | 3 | 0 | 15 |
| *Laboratory technicians* | 5 | 2 | 0 | 3 |
| *Nurses* | 3 | 0 | 0 | 3 |
| *Pharmacists* | 3 | 1 | 0 | 2 |
| *Administration* | 2 | 1 | 0 | 1 |
| *Others* | 6 | 1 | 0 | 5 |
| Soldiers | 22 | 5 | 0 | 17 |
| Sellers | 11 | 4 | 4 | 3 |
| Imams | 3 | 1 | 0 | 2 |
| Journalists | 3 | 0 | 0 | 3 |

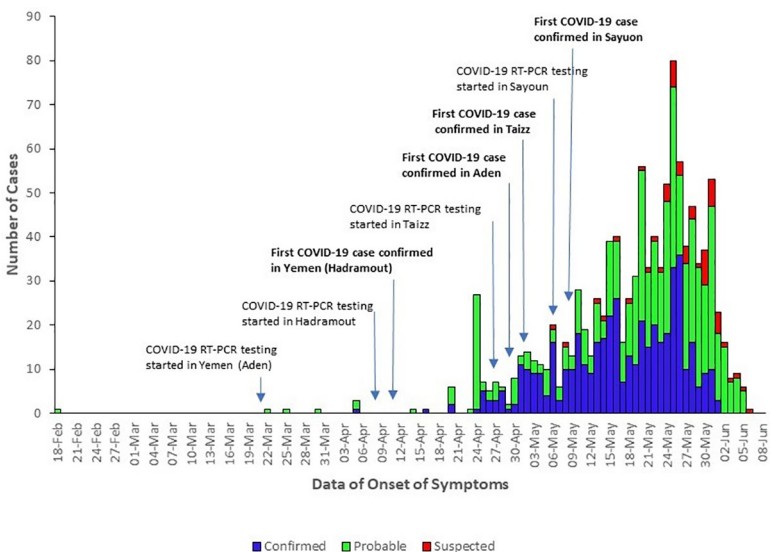

**Fig 2. Distribution of suspected, probable and confirmed cases of SARS-CoV-2 infection in Yemen by date of onset, from 18 April to 5 June 2020.**

recorded in the line list. Of these, 73% (n = 32) died within 24 hours after admission. The mean time from admission to death was 1.1 days. The mean time from admission to death and from onset of symptoms to death per district is described in the S1 Table. The date of death was missing in 51 cases (S1 Table). Of all the fatal cases reported, 6 individuals were reported to be dead on arrival (3 from Mukalla City in Hadramout, 2 from Qairah District in Taizz, and 1 from Salh District in Taizz). Al-Mukalla City in Hadramout Governorate reported 28% (n = 31) of all the deaths in the country (S2 Table) (Fig 3).

## Contact tracing and chains of transmission

From 5 April to 25 May 2020, a total of 18 independent chains of transmission were detected during the investigations, including 3 chains of transmission detected through contact tracing in Hadramout Governorate. One chain of transmission in Hadramout remained active on 25 May 2020, with contacts under follow-up. The chains of transmission generated a total of 33 cases, excluding the index case, of which 18 (55%) were household contacts. The source of infection of the index cases was unknown in 14 chains of transmission, while history of travel to Aden Governorate (n = 3) and history of travel to Saudi Arabia (n = 1) within 14 days before the onset of symptoms was reported by the index case in the other 4 chains of transmission (Fig 4). All the chains of transmission had 1 generation of cases, except one which had 2 generations. The average number of cases generated by each index case was 2 (range: 1–5). The average time between the onset of symptoms of the index case to onset of symptoms of the contacts was 8.8 days (range: 2–22 days). The source of infection and the cases generated by 180 cases were not identified. The contact tracing activities were interrupted in Aden Governorate in the first week of the epidemic due to limited human resource capacity, community resistance, and security issues.

## Discussion

The first 2 months after confirmation of the SARS-CoV-2 epidemic in Yemen was characterized by a 57% hospitalization rate in the southern and eastern parts of the country included in

**Table 5. Distribution of confirmed cases and deaths of SARS-CoV-2 infections by place of treatment in Yemen, 18 February to 05 June 2020.**

|  | Cases | Deaths | Recovered |
|---|---|---|---|
|  | N (%) | N (%) | N (%) |
| Total | 469 | 111 | 23 |
| Home Isolation | 201 (43) | 16 (14) | 4 (17) |
| Admission | 268 (57) | 95 (86) | 17 (74) |
| ICU admission | 27 (6) | 14 (13) | 2 (9) |
| Use of Ventilator | 11 (2) | 8 (7) | 0 (0) |

our study, 63% of deaths occurring in individuals aged <60 years, confirmatory testing of <50% of the suspected cases, and majority of cases were not related to a defined chain of transmission. Therefore, the country satisfied the criteria to be classified as a community transmission setting [12]. The percentage of severe cases requiring admission in Yemen was more than the double that reported in several other countries including Iran, where the percentage was reported to be approximately 20% [1, 5, 13, 14]. However, the mean time from onset of symptoms to admission to the health facility was 5.4 days, which is comparable to the average of 5–10 days reported in other countries [1, 15–18].

Closure of some hospitals, and other hospitals refusing to receive and admit patients with acute respiratory illness may had led to notification of more serious cases, as cases of mild and moderate severity were not self-reported [19]. The closure of health facilities was mainly due to a shortage of personal protective equipment and other supplies to manage SARS-CoV-2 [7]. In addition, there was a shortage of trained RRTs, who were responsible for investigation and collection of laboratory samples of each case reported in the community and in health facilities [19]. This may also have led to underreporting, especially of mild cases that were not self-

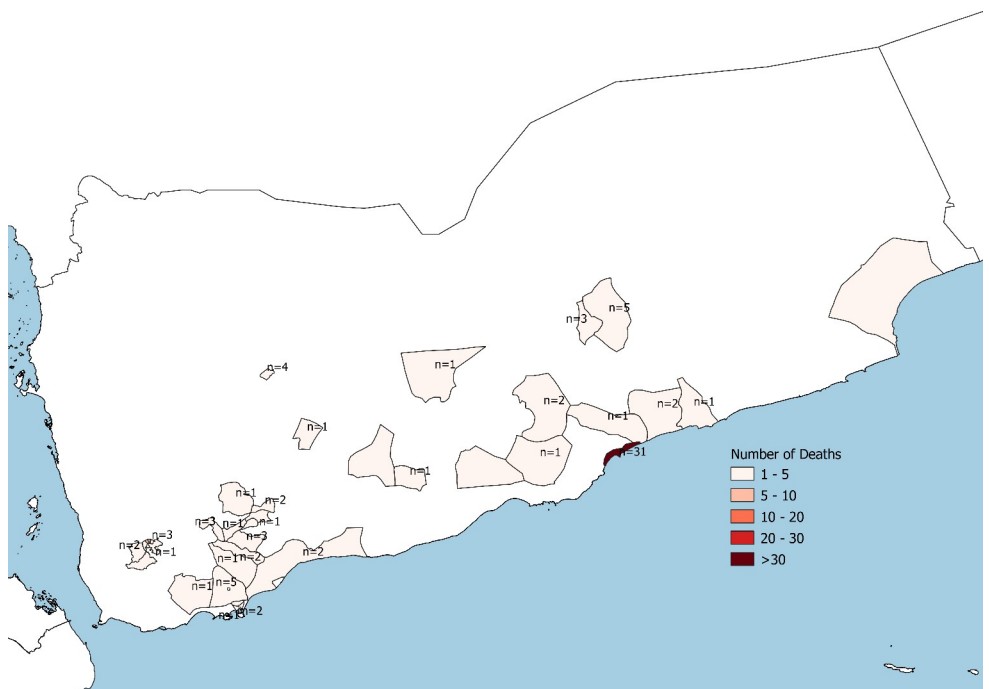

**Fig 3. Geographical location of the confirmed SARS-CoV-2 deaths in Yemen from 18 February to 5 June 2020.**

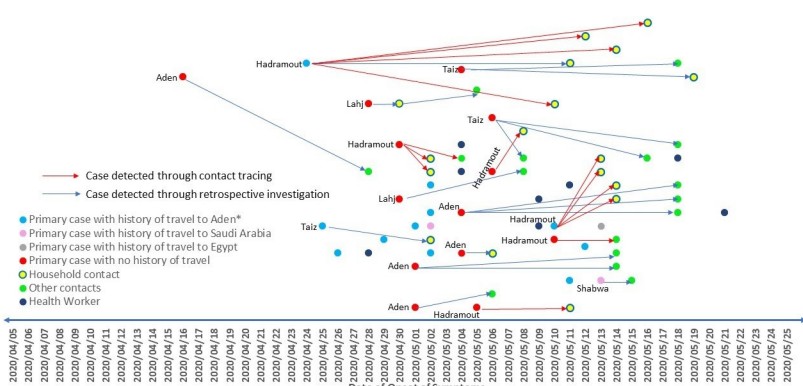

**Fig 4. Chains of transmission among individuals with confirmed SARS-CoV-2 infection identified in Yemen from 5 April to 25 May 2020.**

reported due to limited resources, stigma, and other barriers. More than 50% of the cases were classified as probable due to a shortage of laboratory supplies to investigate all the cases reported [19]. The case fatality rate was also overestimated in China (>20%) at the beginning of the epidemic, when testing capacity was limited, but dramatically decreased once tests became widely available in the country [5]. As Yemen was one of the last countries in the world to notify the first case, the sensitivity of the case definition remained high for long time, capturing mainly individuals with history of travel, and those with severe disease. Therefore, it can be inferred that most of the mild and moderate cases were not reported. The fact that the index case had no clear source of infection and was identified on the day after the laboratory start testing for SARS-CoV-2 suggests that the infection was probably circulating in the country for a period prior to confirmation of the first case. The low prioritization of the component "c" of the case definition may have led to misdiagnosis of severe cases admitted to the hospitals before the confirmation of the first case. The epidemiological curve shows 20 confirmed cases reported in Aden with date of onset of symptoms in the days before the confirmation of the first case and with no clear source of infection. However, it is possible that those who died of the infection may have had other acute and chronic comorbidities besides those captured in the line lists which may have contributed to their deaths [20]. The mean age at death was 54 years, which is much younger than that reported in Iran, China and other countries where the mean age varies from 65 to 73 years [14, 21, 22]. In addition, 63% of all deaths were in individuals aged under 60 years, while other countries have reported less than 20% of deaths in this age group [5]. Considering that the older people are at high risk for severe illness [6], in Yemen many of them likely died at home without reaching the health facilities. Among the individuals who died in the hospitals, the mean time from admission to death was 1.1 days and 73% of deaths occurred within 24 hours of hospital admission. In other countries, in-hospital deaths due to SARS-CoV-2 have generally been reported to occur several days after admission [15, 16, 23, 24]. This indicates that individuals tended to reach health facilities late, with less chance of survival, especially among those who required mechanical ventilation [25]. As in other countries, there was no deaths reported in children aged under 15 years in Yemen [24]. Yemen is experiencing outbreaks of other infectious diseases such as dengue fever, measles, diphtheria, cholera [7, 19], and the H1N1 influenza virus has also been identified in Yemen [26]. Therefore, co-existence of these diseases among SARS-COV-2 infections should be considered. Khat is a plant consumed in few countries, including Yemen as stimulant of central nervous system with effects similar to amphetamines [27]. The majority of adults in Yemen

consume khat, but the effect that this has among individuals with acute and chronic illnesses remains unclear [8, 28]. It has been described in studies conducted in Yemen that khat reduces the appetite, increases blood pressure and induces respiratory diseases [29–31]. However, no study has been conducted to assess the magnitude of these respiratory diseases, and so the possible effect of khat among individuals with SARS-CoV-2 infection is unknown. Malnutrition, especially among adults should also be considered among the comorbidities contributing to SARS-CoV-2 deaths in Yemen [7]. A higher number deaths was also observed in the areas with more limited resources Iran and China in the early stages of SARS-CoV-2 epidemic [23, 32].

The mean age among confirmed cases was 47 years, with higher percentage of cases among males. This is similar that the demographics described in China in the early stage of the epidemic [18, 20]. However, considering that population pyramid in poor countries is usually younger that in developed countries, we also consider underreporting of cases among young people with mild symptoms in Yemen. The signs and symptoms of the cases in Yemen were also similar to those described in China [1, 33, 34]. In Yemen, some patient reported a loss of the sense of taste and smell as described in other countries [35].

The contact tracing was successful in the beginning of the epidemic, especially in Hadramout Governorate, where the index case was detected. However, it was more complex in Aden, where the first cases detected had no clear source of infection, human resources were limited, there was community resistance, and limited supplies, especially PPEs for the contact tracers. In addition, contact tracing in conflict settings may be also a challenge due to security and accessibility issues [36]. However, household contacts, and healthcare workers were among the most affected people, which is similar to what was described in China [17].

Our study has several potential limitations. The findings only include the southern and eastern part of the country, which is home to 31% of the total population of Yemen, as the data from the northern part of the country was inaccessible for analysis. This makes our results not generalizable to the rest of the country, because besides the higher population, the northern part of Yemen has different characteristics including lower temperature and higher altitude. Some studies suggest that different climatic conditions, including temperature and altitude may affect the transmission and mortality due to SARS-CoV-2 infections [37, 38]. The case fatality rate is unclear, as most of the cases reported through the surveillance system were severe. Therefore, we excluded this indicator from our analysis to avoid misinterpretations. The nutrition status of the cases among both adults and children was not reported in the line list, making it impossible to determine the prevalence of malnutrition among SARS-CoV-2 cases. The majority of SARS-CoV-2 articles published with detailed epidemiological analysis are from China and other developed countries. This makes comparisons challenging as Yemen is a country with very different characteristics. The line list used for the analysis had a lot missing information, including key dates and key variables. However, this description of the findings and challenges may be useful for documentation and as a basis for assessing the improvement of the surveillance system in the later stages of the SARS-CoV-2 epidemic in Yemen.

## Conclusions

The surveillance strategy implemented in the first 2 months of the SARS-CoV-2 in Yemen, where 5 RRT members were responsible of investigation of each case in the communities and health facilities was shown to have limited effectiveness, especially in the areas when the available resources were the most limited. The surveillance system in the southern and eastern governorates included in our study captured mainly severe cases, making it difficult to interpret

the mortality data. The mortality appeared to be higher in individuals aged under 60 years, and most fatalities occurred in individuals who were in critical condition when they reached the health facilities. It is unclear whether the presence of other acute comorbidities contributed to the high death rate among SARS-CoV-2 cases in Yemen.

We recommend a revision of the surveillance strategy, to reduce the burden on the RRTs and the investigation of additional acute comorbidities among SARS-CoV-2 patients in Yemen, including H1N1, dengue fever, malnutrition, and the effects of khat.

## Supporting information

**S1 Table. Time from onset of symptoms to admission and deaths due to confirmed SARS-CoV-2 infection in Yemen, 18 February to 05 June 2020.**
(DOCX)

**S2 Table. The districts reporting the higher number of confirmed deaths due to SARS-CoV-2 in Yemen, 18 February to 05 June 2020.**
(DOCX)

## Acknowledgments

We thank all the rapid response teams' members across the country who conducted the investigations of the cases under difficult circumstances. We also thank the health workers and the partners who support the response activities in the country.

## Author Contributions

**Conceptualization:** Ali Ahmed Al-Waleedi, Jeremias D. Naiene, Ahmed A. K. Thabet, Adham Dandarawe, Mohammed Rajamanar, Riyadh Al-Jariri, Mansoor Al Hyubaishi, Nuha Mahmoud, Roy Cosico, Philip Smith, Altaf Musani.

**Data curation:** Hanan Salem, Lina Khanbari, Najib Thabit.

**Formal analysis:** Jeremias D. Naiene, Adham Dandarawe, Maysa Al Noban, Ammar Shawqi, Mikiko Senga.

**Investigation:** Hanan Salem, Nagat Mohammed, Maysa Al Noban, Nasreen Salem Bin-Azoon, Lina Khanbari, Najib Thabit, Basel Obaid, Manal Baaees, Denise Assaf.

**Methodology:** Ali Ahmed Al-Waleedi, Jeremias D. Naiene, Ahmed A. K. Thabet, Adham Dandarawe, Lina Khanbari, Najib Thabit, Denise Assaf, Mikiko Senga, Ismail Mahat Bashir, Philip Smith.

**Supervision:** Adham Dandarawe, Hanan Salem, Mohammed Rajamanar, Riyadh Al-Jariri, Mansoor Al Hyubaishi, Nuha Mahmoud, Roy Cosico, Philip Smith, Altaf Musani.

**Visualization:** Jeremias D. Naiene, Ammar Shawqi.

**Writing – original draft:** Jeremias D. Naiene.

**Writing – review & editing:** Ali Ahmed Al-Waleedi, Jeremias D. Naiene, Ahmed A. K. Thabet.

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
