## [Decision Letter · Decision Letter 0]

10 Sep 2020

PONE-D-20-23480

The First 2 Months of the SARS-CoV-2 Epidemic in Yemen: Analysis of the Surveillance Data

PLOS ONE

Dear Dr. Naiene,

Thank you for submitting your manuscript to PLOS ONE. After careful consideration, we feel that it has merit but does not fully meet PLOS ONE’s publication criteria as it currently stands. Therefore, we invite you to submit a revised version of the manuscript that addresses the points raised during the review process.

ACADEMIC EDITOR: I have received the comments of the reviewers on your manuscript. The specific comments of the reviewers are included below. Please provide point by point response in your revised manuscript.

We look forward to receiving your revised manuscript.

Kind regards,

Muhammad Adrish

Academic Editor

PLOS ONE

Journal Requirements:

Reviewers' comments:

Reviewer's Responses to Questions

**Comments to the Author**

1. Is the manuscript technically sound, and do the data support the conclusions?

Reviewer #1: Yes

Reviewer #2: No

2. Has the statistical analysis been performed appropriately and rigorously? 

Reviewer #1: Yes

Reviewer #2: No

3. Have the authors made all data underlying the findings in their manuscript fully available?

Reviewer #1: Yes

Reviewer #2: Yes

4. Is the manuscript presented in an intelligible fashion and written in standard English?

Reviewer #1: Yes

Reviewer #2: Yes

5. Review Comments to the Author

Reviewer #1: The authors describe Yemen’s experience with COVID19 with an epidemiologic lens. This important work contributes to our understanding of the negative health impacts that COVID19 can have in low-resource settings, and this pandemic compounds other health and humanitarian issues in Yemen. Please consider the following minor and major suggestions.

Minor:

Line 107 – The Case Fatality Rate tends to have a range. These may be good reference to add as well:

1. https://www.thelancet.com/journals/laninf/article/PIIS1473-3099(20)30244-9/fulltext

2. https://www.who.int/news-room/commentaries/detail/estimating-mortality-from-covid-19

Introduction is long. I would remove some of the sentences re: COVID19 and focus on the Yemen component (as was done in the latter half).

Major:

Line 310 discussing underreporting of mild cases, and this is most certainly accurate. Much more expansion on this concept is warranted. Due to limited resources and many barriers to care, there is the potential for tremendous selection bias here which should be acknowledged. This is of no fault of the authors, but represents a biased sample. The authors are asked to add this to their discussion.

Related to the point above - Many at risk for severe illness (e.g. those over the age of 60) will die at home and never make it to the hospital, which could explain why deaths are primarily recorded in those in their 50s. The authors are asked to add this to their discussion.

Also related to the first point, the demographics of Yemen are young. The average age is 19. The average age of Germany is 47. Even with the health disparities and malnutrition in Yemen, there is the possibility of major community spread among younger people who are less likely to have severe outcomes due to age, resulting in a vast underreporting of cases. The authors are asked to add this to their discussion.

Reviewer #2: Many thanks for this very useful description of the SARS-CoV-2 epidemic onset on Yemen and recommendations to strengthen the surveillance system are indeed very welcome.

Few clarifications remain to be provided to strengthen or nuance the conclusion of this descriptive study:

- The sample only includes 31% of the total population. The generalisability of the results should be reviewed and conclusions nuanced.

- Contact tracing is available from only one district. Contact tracing interruption and inconsistence from May 25th reported. More information on the inconsistencies and control that these inconsistencies were not present before May 25th should be provided

- Component C of the case definition ‘was given low priority for several weeks. What and when was the prioritisation of these cases changed? A justification and impact on results needs to be reported.

- Patients mean age range is reported, with a range from 1 to 90 years old. Given the age profile of SARS-CoV-2 patients, the median age might be more relevant.

- Some results are shown from Feb 18th and some only from Apr 5th to June 5th. The potential bias involved by a possible early analysis is to be explored. Justification of the timeframe analysis is to be provided. Was a difference observed if the same analysis was performed from Feb 18th or from Apr 5th and is the difference statistically significant?

- The comparison with other countries is not statistically supported by the results, but rather mentioned in the discussion. However, it is part of the overall conclusion of the study. Comparison with other countries should be done more thoroughly if included in the overall conclusion of the study.

6. PLOS authors have the option to publish the peer review history of their article (what does this mean?). If published, this will include your full peer review and any attached files.

Reviewer #1: No

Reviewer #2: No

---

## [Author Response · Author response to Decision Letter 0]

13 Sep 2020

Response to the comments made by Reviewer 1

“Line 107 – The Case Fatality Rate tends to have a range. These may be good reference to add as well:

1. https://www.thelancet.com/journals/laninf/article/PIIS1473-3099(20)30244-9/fulltext

2. https://www.who.int/news-room/commentaries/detail/estimating-mortality-from-covid-19”

Response: Many thanks for the suggestions. We have included the range and the reference following your advice.

“Introduction is long. I would remove some of the sentences re: COVID19 and focus on the Yemen component (as was done in the latter half)”

Response: Thank you for your advice. The introduction had 483 words and following your recommendation we have removed some of the background information that were less relevant. We have now 434 words with more focus on Yemen component. 

“Line 310 discussing underreporting of mild cases, and this is most certainly accurate. Much more expansion on this concept is warranted. Due to limited resources and many barriers to care, there is the potential for tremendous selection bias here which should be acknowledged. This is of no fault of the authors, but represents a biased sample. The authors are asked to add this to their discussion.”

Response: We understand that the underreporting of mild cases required more expansion. Therefore, we extensively described it in the previous paragraph as following: “Closure of some hospitals, and other hospitals refusing to receive and admit patients with acute respiratory illness may had led to notification of more serious cases, as cases of mild and moderate severity were not self-reported (18). The closure of health facilities was mainly due to a shortage of personal protective equipment and other supplies to manage SARS CoV 2 (6). In addition, there was a shortage of trained RRTs, who were responsible for investigation and collection of laboratory samples of each case reported in the community and in health facilities (18). This may also have led to underreporting, especially of mild cases that were not self-reported”. Following your advice, we added to our discussion the underreporting of mild cases due to limited resources, stigma, and other barriers. 

“Related to the point above - Many at risk for severe illness (e.g. those over the age of 60) will die at home and never make it to the hospital, which could explain why deaths are primarily recorded in those in their 50s. The authors are asked to add this to their discussion.”

Response: Thank you for this excellent comment. We added it to our discussion as advised. 

Also related to the first point, the demographics of Yemen are young. The average age is 19. The average age of Germany is 47. Even with the health disparities and malnutrition in Yemen, there is the possibility of major community spread among younger people who are less likely to have severe outcomes due to age, resulting in a vast underreporting of cases. The authors are asked to add this to their discussion.

Response: Thank you for this particularly good observation also. We added the comment in our discussion as advised. 

Response to the comments made by Reviewer 2

“- The sample only includes 31% of the total population. The generalisability of the results should be reviewed and conclusions nuanced.”

Response: We agree that our findings cannot be generalized to the rest of the country. Therefore, We have clarified the generalizability of our findings in the discussion and conclusion sections as advised. 

“- Contact tracing is available from only one district. Contact tracing interruption and inconsistence from May 25th reported. More information on the inconsistencies and control that these inconsistencies were not present before May 25th should be provided”

Response: Thank you for this observation. We have provided more information regarding the contact tracing and the limitation of the contacts to a maximum of 10 contacts per team to ensure the consistence and quality of contact tracing. The inconsistence on data collection was due to increase of contacts compared to the contact tracing team as we clarified in the manuscript. 

“- Component C of the case definition ‘was given low priority for several weeks. What and when was the prioritisation of these cases changed? A justification and impact on results needs to be reported.”

Response: The Ministry of health and the health facilities decided to give more attention to the component “c” of the case definition when the hospitals noticed an increase in admissions of cases of severe acute respiratory illnesses. However, we were unable to determine the exact time as it was not documented by the Ministry of health. The impact was that more cases of severe acute respiratory illnesses were investigated for COVID-19 leading to detection of the first case in country, which had no history of travel or contact with sick people as stated in the component “c” of the case definition. We have reported in the discussion section the impact of the low prioritization of the component “c” of the case definition as advised. 

“- Patients mean age range is reported, with a range from 1 to 90 years old. Given the age profile of SARS-CoV-2 patients, the median age might be more relevant.”

Response: Thank you for this suggestion. We have replaced the mean age (47 years) by the median age (46 years)

“- Some results are shown from Feb 18th and some only from Apr 5th to June 5th. The potential bias involved by a possible early analysis is to be explored. Justification of the timeframe analysis is to be provided. Was a difference observed if the same analysis was performed from Feb 18th or from Apr 5th and is the difference statistically significant?”

Response: Many thanks for this observation. 5 April was the date of onset of the first confirmed case in Yemen. Therefore, some results reported from 5 April refer only to confirmed cases. This included the Table 5, and Figure 3. To make it clear to the readers we have made all the tables and figures consistent, with results from 18 February to 5 June, specifying the ones reporting only confirmed cases and the ones reporting also the suspected and probable cases. We left the results from 4 April in the contact tracing and chains of transmission analysis but we have clarified in the methods section that the contact tracing data collection started with the first case who had the date of onset on 5 April 2020.

“- The comparison with other countries is not statistically supported by the results, but rather mentioned in the discussion. However, it is part of the overall conclusion of the study. Comparison with other countries should be done more thoroughly if included in the overall conclusion of the study.”

Response: Thank you for this point. We have removed from our overall conclusions the comparison with other countries.

---

## [Decision Letter · Decision Letter 1]

22 Sep 2020

PONE-D-20-23480R1

The First 2 Months of the SARS-CoV-2 Epidemic in Yemen: Analysis of the Surveillance Data

PLOS ONE

Dear Dr. Naiene,

Thank you for submitting your manuscript to PLOS ONE. After careful consideration, we feel that it has merit but does not fully meet PLOS ONE’s publication criteria as it currently stands. Therefore, we invite you to submit a revised version of the manuscript that addresses the points raised during the review process.

ACADEMIC EDITOR: I have received the comments of the reviewers on your manuscript. The specific comments of the reviewers are included below. Please provide point by point response in your revised manuscript.

We look forward to receiving your revised manuscript.

Kind regards,

Muhammad Adrish

Academic Editor

PLOS ONE

Reviewers' comments:

Reviewer's Responses to Questions

**Comments to the Author**

1. If the authors have adequately addressed your comments raised in a previous round of review and you feel that this manuscript is now acceptable for publication, you may indicate that here to bypass the “Comments to the Author” section, enter your conflict of interest statement in the “Confidential to Editor” section, and submit your "Accept" recommendation.

Reviewer #1: All comments have been addressed

Reviewer #2: All comments have been addressed

2. Is the manuscript technically sound, and do the data support the conclusions?

Reviewer #1: Yes

Reviewer #2: Yes

3. Has the statistical analysis been performed appropriately and rigorously? 

Reviewer #1: Yes

Reviewer #2: Yes

4. Have the authors made all data underlying the findings in their manuscript fully available?

Reviewer #1: Yes

Reviewer #2: Yes

5. Is the manuscript presented in an intelligible fashion and written in standard English?

Reviewer #1: Yes

Reviewer #2: Yes

6. Review Comments to the Author

Reviewer #1: The authors have satisfactorily addressed reviewers' issues.

While there were not overt methodological limitations, there was limited data available and biased sample collection beyond the control of study staff, however interpretation of the limited data available has now been re-framed to reflect weaknesses in data collection and is presented in the appropriate context.

Reviewer #2: Very minor comments

- Abstract conclusion: Authors are invited to present their results within the sampling frame of the Southern and Eastern part of Yemen.

Line 78 : Authors should acknowledge the potential sampling and under-reporting impact on the results to nuance the picture of the two first months

Line 219: Authors are invited to explain and add a reference to table S1

Line 282-287: Authors should state that the presented descriptive statistics are representative of the southern and Eastern part.

Line 359-360: The reference to high altitude and low temperature could lead to think the two variables could have a possible effect on COVID-19 transmission and profile. This should be referenced or edited accordingly.

7. PLOS authors have the option to publish the peer review history of their article (what does this mean?). If published, this will include your full peer review and any attached files.

Reviewer #1: No

Reviewer #2: No

---

## [Author Response · Author response to Decision Letter 1]

22 Sep 2020

Response to the comments made by Reviewer 1

“While there were not overt methodological limitations, there was limited data available and biased sample collection beyond the control of study staff, however interpretation of the limited data available has now been re-framed to reflect weaknesses in data collection and is presented in the appropriate context.”

Response: Many thanks for your time and the valuable comments that improved our manuscript. 

Response to the comments made by Reviewer 2

“- Abstract conclusion: Authors are invited to present their results within the sampling frame of the Southern and Eastern part of Yemen.”

Response: Thank you for the comments. We have specified in the abstract conclusion the sampling frame of the southern and eastern governorates. 

“Line 78 : Authors should acknowledge the potential sampling and under-reporting impact on the results to nuance the picture of the two first months”

Response: Many thanks for this observation. We totally agree that the potential sampling and under-reporting impact on the results should be acknowledged. It is extensively mentioned in the discussions and limitations of our study. Now we have included also in the abstract as suggested

“Line 219: Authors are invited to explain and add a reference to table S1”

Response: Thank you very much for this suggestion. The reference to table S1 is already in line 251. We have included the explanation of the table as suggested and additional reference in the results sections. In addition, we have corrected a minor error in the totals on the same S1 table that does not affected the results and conclusions in the manuscript. 

“Line 282-287: Authors should state that the presented descriptive statistics are representative of the southern and Eastern part.”

Response: Many thanks again for this observation. We have included the statement as suggested

“Line 359-360: The reference to high altitude and low temperature could lead to think the two variables could have a possible effect on COVID-19 transmission and profile. This should be referenced or edited accordingly.”

Response: Thank you for this excellent observation. We have included two references of effects of temperature and altitude on COVID-19 transmission profile.

---

## [Decision Letter · Decision Letter 2]

13 Oct 2020

The First 2 Months of the SARS-CoV-2 Epidemic in Yemen: Analysis of the Surveillance Data

PONE-D-20-23480R2

Dear Dr. Naiene,

We’re pleased to inform you that your manuscript has been judged scientifically suitable for publication and will be formally accepted for publication once it meets all outstanding technical requirements.

Kind regards,

Muhammad Adrish

Academic Editor

PLOS ONE

Additional Editor Comments (optional):

Reviewers' comments:

Reviewer's Responses to Questions

**Comments to the Author**

1. If the authors have adequately addressed your comments raised in a previous round of review and you feel that this manuscript is now acceptable for publication, you may indicate that here to bypass the “Comments to the Author” section, enter your conflict of interest statement in the “Confidential to Editor” section, and submit your "Accept" recommendation.

Reviewer #2: All comments have been addressed

2. Is the manuscript technically sound, and do the data support the conclusions?

Reviewer #2: Yes

3. Has the statistical analysis been performed appropriately and rigorously? 

Reviewer #2: Yes

4. Have the authors made all data underlying the findings in their manuscript fully available?

Reviewer #2: Yes

5. Is the manuscript presented in an intelligible fashion and written in standard English?

Reviewer #2: Yes

6. Review Comments to the Author

Reviewer #2: Many thanks for addressing all the comments and for this very informative study,

A final suggestion to not undermine the overall conclusion of this valuable study:

line 75: could be edited by mentioning that within the scope of the study, 'the mild and moderate cases...'. this would allow to remove the 81 to 83 to not finish the conclusion with a limitation

7. PLOS authors have the option to publish the peer review history of their article (what does this mean?). If published, this will include your full peer review and any attached files.

Reviewer #2: No

---

## [Editor Report · Acceptance letter]

16 Oct 2020

PONE-D-20-23480R2 

The First 2 Months of the SARS-CoV-2 Epidemic in Yemen: Analysis of the Surveillance Data 

Dear Dr. Naiene:

I'm pleased to inform you that your manuscript has been deemed suitable for publication in PLOS ONE. Congratulations! Your manuscript is now with our production department. 

Kind regards, 

on behalf of

Dr. Muhammad Adrish 

Academic Editor

PLOS ONE